# Rate Constants and Branching Ratios for the Self-Reaction of Acetyl Peroxy (CH$_3$C(O)O$_2$$^\bullet$) and Its Reaction with CH$_3$O$_2$

Mohamed Assali and Christa Fittschen *

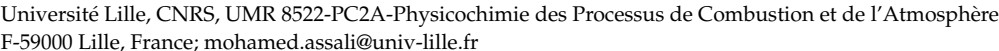

Université Lille, CNRS, UMR 8522-PC2A-Physicochimie des Processus de Combustion et de l'Atmosphère, F-59000 Lille, France; mohamed.assali@univ-lille.fr
* Correspondence: christa.fittschen@univ-lille.fr

**Abstract:** The self-reaction of acetylperoxy radicals (CH$_3$C(O)O$_2$$^\bullet$) (R1) as well as their reaction with methyl peroxy radicals (CH$_3$O$_2$$^\bullet$) (R2) have been studied using laser photolysis coupled to a selective time resolved detection of three different radicals by cw-CRDS in the near-infrared range: CH$_3$C(O)O$_2$$^\bullet$ was detected in the Ã-$\tilde{X}$ electronic transition at 6497.94 cm$^{-1}$, HO$_2$$^\bullet$ was detected in the 2$\nu_1$ vibrational overtone at 6638.2 cm$^{-1}$, and CH$_3$O$_2$$^\bullet$ radicals were detected in the Ã-$\tilde{X}$ electronic transition at 7489.16 cm$^{-1}$. Pulsed photolysis of different precursors at different wavelengths, always in the presence of O$_2$, was used to generate CH$_3$C(O)O$_2$$^\bullet$ and CH$_3$O$_2$$^\bullet$ radicals: acetaldehyde (CH$_3$CHO/Cl$_2$ mixture or biacetyl (CH$_3$C(O)C(O)CH$_3$) at 351 nm, and acetone (CH$_3$C(O)CH$_3$) or CH$_3$C(O)C(O)CH$_3$ at 248 nm. From photolysis experiments using CH$_3$C(O)C(O)CH$_3$ or CH$_3$C(O)CH$_3$ as precursor, the rate constant for the self-reaction was found with $k_1$ = (1.3 ± 0.3) × 10$^{-11}$ cm$^3$s$^{-1}$, in good agreement with current recommendations, while the rate constant for the cross reaction with CH$_3$O$_2$$^\bullet$ was found to be $k_2$ = (2.0 ± 0.4) × 10$^{-11}$ cm$^3$s$^{-1}$, which is nearly two times faster than current recommendations. The branching ratio of (R2) towards the radical products was found at 0.67, compared with 0.9 for the currently recommended value. Using the reaction of Cl$^\bullet$-atoms with CH$_3$CHO as precursor resulted in radical profiles that were not reproducible by the model: secondary chemistry possibly involving Cl$^\bullet$ or Cl$_2$ might occur, but could not be identified.

**Keywords:** peroxy radicals; acetyl peroxy; laser photolysis; cavity ring down spectroscopy

## 1. Introduction

The oxidation of volatile organic compounds (VOCs) in the troposphere is mainly driven by hydroxyl radicals ($^\bullet$OH) and leads, after the addition of O$_2$, to the formation of organic peroxy radicals (RO$_2$$^\bullet$). The fate of these RO$_2$$^\bullet$ radicals depends on the chemical composition of the environment: in a polluted atmosphere they react mainly with nitric oxide (NO) to form alkoxy radicals or react with nitrogen dioxide (NO$_2$) to form peroxynitrates (RO$_2$NO$_2$). Subsequent to the reaction with NO, alkoxy radicals react with O$_2$ to form hydroperoxy radicals (HO$_2$$^\bullet$). HO$_2$$^\bullet$ further oxidises NO into NO$_2$ and thus regenerates $^\bullet$OH, closing the quasi-catalytic cycle. The subsequent photolysis of produced NO$_2$ is the only relevant chemical source of tropospheric ozone. In clean environments with low NO$_x$ (NO$_x$ = NO + NO$_2$) concentrations, the dominant loss of RO$_2$$^\bullet$ is due to its reaction with HO$_2$$^\bullet$ forming hydroperoxides ROOH and terminating the radical reaction chain. In addition, RO$_2$$^\bullet$ radicals can react either with other RO$_2$$^\bullet$ as self-(RO$_2$$^\bullet$ + RO$_2$$^\bullet$) or cross-reaction (RO$_2$$^\bullet$ + R'O$_2$$^\bullet$), or with $^\bullet$OH radicals (RO$_2$$^\bullet$ + $^\bullet$OH) [1–4].

The majority of emitted biogenic non-methane hydrocarbons are isoprene (53%) and monoterpene species (16%) [5]. The photooxidation of these highly abundant compounds and their oxidation products form among other products significant amounts of acetylperoxy radicals (CH$_3$C(O)O$_2$$^\bullet$). In the reaction with NO$_2$, CH$_3$C(O)O$_2$$^\bullet$ form peroxyacetyl nitrate (PAN), which is a toxic secondary air pollutant. In addition, PAN acts as the principal tropospheric reservoir species for NO$_x$ [6]. The only relevant source in the troposphere

is this photochemical process, so that PAN is an indicator for photochemical oxidation. Its relatively long atmospheric lifetime of approximately two weeks allows for transport over long distances.

Model calculations of measured radical concentrations in different field studies underestimate $HO_x^{\bullet}$ ($HO_x^{\bullet} = {}^{\bullet}OH + HO_2^{\bullet}$) radical concentrations in remote regions with high emissions of VOCs from biogenic sources [7–10]. Even though instrumental interferences might be partially responsible for this underestimation by models [11,12], unidentified chemistry, erroneous rate constants, or branching ratios of key reactions might play an important role, too. Because acetylperoxy radicals are formed from biogenic precursors and serve as source for $HO_2^{\bullet}$, understanding their properties under low $NO_x$ conditions is of importance.

Major pathways for $CH_3C(O)O_2^{\bullet}$ under low $NO_x$ are

- its self-reaction

$$2\ CH_3C(O)O_2^{\bullet} \rightarrow 2\ CH_3C(O)O^{\bullet} + O_2 \tag{R1}$$

- its reaction with $CH_3O_2^{\bullet}$

$$CH_3C(O)O_2^{\bullet} + CH_3O_2^{\bullet} \rightarrow CH_3O^{\bullet} + CH_3C(O)O^{\bullet} + O_2 \tag{R2a}$$

$$\rightarrow CH_3C(O)OH + HCHO + O_2 \tag{R2b}$$

whereby (R2a) maintains the radical pool, while (R2b) is a termination reaction. Therefore, a reliable determination of the branching ratio is of importance.

- its reaction with $HO_2$

$$CH_3C(O)O_2^{\bullet} + HO_2^{\bullet} \rightarrow CH_3C(O)OOH + O_2 \tag{R3a}$$

$$\rightarrow CH_3C(O)OH + O_3 \tag{R3b}$$

$$\rightarrow CH_3C(O)O^{\bullet} + {}^{\bullet}OH + O_2 \tag{R3c}$$

The first two pathways lead to radical chain terminating products (R3a) and (R3b), while the third path also regenerates $^{\bullet}OH$ (R3c).

Investigation of the $CH_3C(O)O_2^{\bullet}$ reaction kinetics is not straight forward, because secondary chemistry cannot be avoided: $CH_3C(O)O^{\bullet}$, the product of the $CH_3C(O)O_2^{\bullet}$ self-reaction (R1), rapidly decomposes and leads, after the addition of $O_2$, to the formation of $CH_3O_2^{\bullet}$ radicals:

$$CH_3C(O)O^{\bullet} \rightarrow CH_3^{\bullet} + CO_2 \tag{R4}$$

$$CH_3^{\bullet} + O_2\ (+M) \rightarrow CH_3O_2^{\bullet}\ (+M) \tag{R5}$$

Given that the rate constant of both reactions (R1) and (R2) are on the same order of magnitude, the $CH_3C(O)O_2^{\bullet}$ decay is thus accelerated by (R2). To make things even more complex, (R2) has two pathways, one of which recycles $CH_3O_2^{\bullet}$ through (R4) and (R5) and simultaneously generates $HO_2^{\bullet}$ radicals through (R6):

$$CH_3O^{\bullet} + O_2 \rightarrow CH_2O + HO_2^{\bullet} \tag{R6}$$

These $HO_2^{\bullet}$ radicals will in turn react with $CH_3C(O)O_2^{\bullet}$, making it hard to distinguish between all reactions. This is even more true, as most of the former studies have been carried out by flash photolysis coupled to a rather unselective detection of the different peroxy radicals involved in the mechanism by UV absorption spectroscopy: the spectral overlap of different peroxy species in this region is prone to systematic errors in the quantitative detection [13–17]. Therefore, experiments quantifying different $RO_2^{\bullet}$ radicals by UV absorption are difficult to evaluate.

(R3) has been studied several times [18–27], especially since the discovery of the radical maintaining channel (R3c) in 2004 by Hasson et al. [26]. Good agreement on the rate constant and on the branching ratio has now been found. (R1) and (R2) have

been studied less often, three times each and always using UV absorption spectroscopy. A good agreement on the rate constant for the $CH_3C(O)O_2{}^{\bullet}$ self-reaction (R1) has been found in the three studies [14,16,17]. The same is true for (R2): a good agreement for the overall rate constant has been found in the three studies [16,17,28], which however must be fortuitous, as the authors used very different branching ratios for (R2a), ranging from 0 [17] over 0.65 [28] to 0.9 [16] for $k_{2a}/k_2$. These different branching ratios for (R2) were also used by the authors for the extraction of $k_1$, so the agreement in $k_1$ must also be fortuitous. A semi-empirical study on the rate constants of self- and cross-reactions of peroxy radicals, based on the calculated stabilisation energy of the tetroxide intermediate, predicts $1.4 \times 10^{-11}$ cm$^3$s$^{-1}$ for (R1) and $7 \times 10^{-12}$ cm$^3$s$^{-1}$ for (R2), which is in good agreement with experiments for (R1) and at the lower end for (R2) [29]. A summary of the available literature data together with the current recommendation by IUPAC [30] (which is very similar to the recommendation by JPL [31]) is given in Table 1.

**Table 1.** Summary of literature results for (R1) and (R2), all rate constants in cm$^3$s$^{-1}$.

| References / Reaction | IUPAC [30] | Roehl et al. [16] | Maricq, Szente [17] | Villenave, Lesclaux [28] | This work |
|---|---|---|---|---|---|
| $CH_3C(O)O_2{}^{\bullet} + CH_3O_2{}^{\bullet}$ | $1.1 \times 10^{-11}$ | $9.8 \times 10^{-12}$ | $1.0 \times 10^{-11}$ | $8.6 \times 10^{-12}$ | $2.0 \times 10^{-11}$ |
| (R2a): $CH_3O^{\bullet} + CH_3C(O)O^{\bullet} + O_2$ | $9.9 \times 10^{-12}$ | $8.8 \times 10^{-12}$ | $0$ | $5.3 \times 10^{-12}$ | $1.3 \times 10^{-11}$ |
| (R2b): $CH_3C(O)OH + HCHO + O_2$ | $1.1 \times 10^{-12}$ | $1.0 \times 10^{-12}$ | $1 \times 10^{-11}$ | $2.9 \times 10^{-12}$ | $6.5 \times 10^{-12}$ |
| $\alpha$ (R2a)/(R2) | $0.9$ | $0.9$ | $0$ | $0.65$ | $0.67$ |

| References / Reaction | IUPAC [30] | Roehl et al. [16] | Maricq, Szente [17] | Moortgat et al. [14] | This work |
|---|---|---|---|---|---|
| $2\,CH_3C(O)O_2{}^{\bullet} \rightarrow 2\,CH_3C(O)O^{\bullet} + O_2$ | $1.6 \times 10^{-11}$ | $1.36 \times 10^{-11}$ | $1.5 \times 10^{-11}$ | $1.6 \times 10^{-11}$ | $1.3 \times 10^{-11}$ |

Given the strong disagreement of the spare literature data, it seems important to investigate reaction (R1) and (R2) again using more selective detection methods. Here, $CH_3C(O)O_2{}^{\bullet}$ and $CH_3O_2{}^{\bullet}$ radicals were detected by absorption in the $\tilde{A}$–$\tilde{X}$ electronic transition, and the $HO_2{}^{\bullet}$ radical by absorption in the $2\nu_1$ overtone vibration band, all located in the near infrared region. To make up for the much smaller absorption cross section in this range compared to the UV range, cavity ring down spectroscopy (cw-CRDS) is used.

## 2. Experimental

### 2.1. Experimental Setup

The setup has been described in detail before [32–37] and is only briefly discussed here. The setup consisted of a 0.79 m long flow reactor made of stainless steel. The beam of a pulsed excimer laser (Lambda Physik LPX 202i) passed the reactor longitudinally. The flow reactor contained two identical continuous wave cavity ring-down spectroscopy (cw-CRDS) absorption paths, which were installed in a small angle with respect to the photolysis path. An overlap with the photolysis beam of 0.288 m is achieved with an excimer beam width delimited to 2 cm. Both beam paths were tested for a uniform overlap with the photolysis beam before experiments were performed. For this purpose, both cw-CRDS instruments were operated to simultaneously measure $HO_2{}^{\bullet}$ concentrations. Deviations between $HO_2{}^{\bullet}$ concentrations were less than 5%, demonstrating that the photolysis laser was well aligned, i.e., both light paths probed a very similar photolysed volume in the reactor. A small helium purge flow prevented the mirrors from being contaminated. Three different distributed feedback (DFB) lasers are used for the detection of the three species ($CH_3C(O)O_2{}^{\bullet}$: Alcatel A1905LMI 3CN004 1 0CR, $6497 \pm 18$ cm$^{-1}$, $HO_2{}^{\bullet}$; NEL NLK1E5GAAA, $6629 \pm 17$ cm$^{-1}$; $CH_3O_2{}^{\bullet}$: NEL NLK1B5EAAA, $7480 \pm 20$ cm$^{-1}$). They are coupled into one of the cavities by systems of lenses and mirrors. Each probe beam passed an acousto-optic modulator (AOM, AAoptoelectronic, Orsay, France) to rapidly turn off the 1st order beam once a threshold for light intensity in the cavity was reached, in order to measure the ring-down event. One

of the cavity mirrors is glued onto a piezo-transducer, which periodically modulates the cavity length in order to bring the cavity into resonance with the wavelength of the DFB lasers. The piezo is controlled by a homemade tracking system [38]. Then, the decay of light intensity is recorded by a fast 16-bit analogue acquisition card (PCI-6259, National Instruments) in a PC. The acquisition card has an acquisition frequency of 1.25 MHz and thus the ring-down signal is sampled every 800 ns and the data are transferred to the PC in real time via PCI bus. An exponential fit is applied to retrieve the ring-down time. Through synchronisation with the trigger of the photolysis laser, the delay between the photolysis pulse and the random occurrence of the ring-down event is registered [37]. A typical kinetic decay is obtained by accumulating ring-down events for 50–100 photolysis pulses and consists of several hundred individual ring-down times that have occurred randomly either before or after the photolysis pulse. The absorption coefficient $\alpha$ is derived from Equation (1).

$$\alpha = [A] * \sigma_A = \frac{R_L}{c}\left(\frac{1}{\tau} - \frac{1}{\tau_0}\right) \tag{1}$$

where $\tau$ is the ring-down time with an absorber present (i.e., at a given delay after the photolysis pulse); $\tau_0$ is the ring-down time with no absorber present (i.e., before the photolysis pulse); $\sigma_A$ is the absorption cross section of the absorbing species $A$; $R_L$ is the ratio between cavity length (0.79 m) and effective absorption path (0.288 m); $c$ is the speed of light. Typical ring-down times of the empty cavity were up to 100 μs, corresponding to the reflectivity of the mirrors of 0.99997.

Acetylperoxy radicals were generated from different precursors by either

- pulsed 351 nm photolysis of acetaldehyde ($CH_3CHO$)/$Cl_2$/$O_2$ mixtures:

$$Cl_2 + h\nu_{351\,nm} \rightarrow 2\,Cl^\bullet \tag{R7}$$

$$CH_3CHO + Cl^\bullet \rightarrow CH_3CO^\bullet + HCl \tag{R8}$$

$$CH_3CO^\bullet + O_2\,(+\,M) \rightarrow CH_3C(O)O_2^\bullet\,(+\,M) \tag{R9a}$$

$CH_3CO^\bullet$ can also react with $O_2$ through other pathways: it has been observed [39,40] that its reaction with $O_2$ can also lead to low concentrations of $HO_2^\bullet$ and $^\bullet OH$, depending on the amount of internal energy of $CH_3CO^\bullet$:

$$CH_3CO^\bullet + O_2 \rightarrow {}^\bullet CH_2CO + HO_2{}^\bullet/CH_2C(O)O^\bullet + {}^\bullet OH \tag{R9b}$$

$CH_3CO^\bullet$ might also decompose before reaction with $O_2$:

$$CH_3CO \rightarrow CH_3 + CO \tag{R10}$$

- pulsed 351 nm photolysis of biacetyl ($CH_3C(O)C(O)CH_3$)/$O_2$ mixtures:

$$CH_3C(O)C(O)CH_3 + h\nu_{351\,nm} \rightarrow 2\,CH_3CO^\bullet \tag{R11}$$

  followed by either (R9) or (R10)
- Pulsed 248 nm photolysis of biacetyl ($CH_3C(O)C(O)CH_3$)/$O_2$ mixtures: The same mechanism as above is utilized, but with a higher fraction of subsequent decomposition (R10)
- Pulsed 248 nm photolysis of acetone ($CH_3C(O)CH_3$)/$O_2$ mixtures

$$CH_3C(O)CH_3 + h\nu_{248\,nm} \rightarrow CH_3CO^\bullet + {}^\bullet CH_3 \tag{R12}$$

  followed by (R5), (R9), and (R10).

Using these different precursors thus allows obtaining different ratios of the initial radical concentrations. Table 2 summarizes the quantum yields such as those obtained from fitting the concentration time profiles for the different species for all precursors at

the different wavelengths. It can be seen that the rate of decomposition (R10) is highest following the 248 nm photolysis of $CH_3C(O)C(O)CH_3$ and this precursor is also the one leading to the highest yield of initial $HO_2$.

**Table 2.** Ratio of $CH_3C(O)O_2^{\bullet}$ and $CH_3O_2^{\bullet}$ radicals obtained from different precursors and at different pressures. Last column shows fraction of $CH_3CO$ radicals that lead in collision with $O_2$ to $HO_2$ radicals (R9).

| Reaction | % $CH_3C(O)O_2^{\bullet}$ | % $CH_3O_2^{\bullet}$ | $k_{9b}/k_9$ |
|---|---|---|---|
| $CH_3CHO + Cl^{\bullet}$ | 100 | 0 | 0.007 * |
| $CH_3C(O)C(O)CH_3 + h\nu_{351\,nm}$ 100 Torr | 89 | 11 | 0.014 |
| $CH_3C(O)C(O)CH_3 + h\nu_{248\,nm}$ 100 Torr | 47 | 53 | 0.079 |
| $CH_3C(O)C(O)CH_3 + h\nu_{248\,nm}$ 200 Torr | 50 | 50 | 0.079 |
| $CH_3C(O)CH_3 + h\nu_{248\,nm}$ 100 Torr | 38 | 62 | 0.064 |
| $CH_3C(O)CH_3 + h\nu_{248\,nm}$ 200 Torr | 40 | 60 | 0.064 |

* in 100 Torr $O_2$. Ratio is around 0.02 in 50 Torr helium.

Acetaldehyde and biacetyl were prepared as diluted mixtures in a glass bulb. A small flow of this mixture was added to the main flow through a calibrated flow meter (Brunkhorst, Tylan). Acetone (Sigma Aldrich, France) was added to the mixture by flowing a small fraction of the main flow through a bubbler containing liquid acetone, kept in ice or in a thermostated water bath. All experiments were carried out at 298 K, and most experiments were performed at a total pressure of 100 Torr $O_2$ (Praxair, 4.5). Some experiments were also carried out in 50 Torr helium (Praxair 4.5). The total flow rate was generally 450 cm$^3$ min$^{-1}$, leading to a flow velocity in the cell of 2.3 cm s$^{-1}$ at 100 Torr. The photolysis repetition rate was generally 0.3 Hz, leading to a renewal of the gas mixture within the observation volume every second photolysis shot: occasional experiments were performed at lower photolysis rates or higher total flows to check for any possible influence of remaining reaction products.

*2.2. Quantification of $CH_3C(O)O_2^{\bullet}$*

The relative spectrum has already been measured by Zalyubovsky et al. [41] in a large wavelength range and the absolute absorption cross section of the strongest band at 5582 cm$^{-1}$ has been estimated. In a recent work, our group has determined absolute absorption cross sections in two wavelength ranges from 6094 to 6180 cm$^{-1}$ and from 6420 to 6600 cm$^{-1}$, corresponding to the C(O)O bend and to the OO stretch transition, respectively [42], whereby the cross sections were determined relative to the absorption cross section of $HO_2^{\bullet}$. Based on this work, $CH_3C(O)O_2^{\bullet}$ was quantified at 6497.94 cm$^{-1}$, with an absorption cross section $\sigma_{CH_3C(O)O_2} = 3.3 \times 10^{-20}$ cm$^2$. The spectrum of $CH_3C(O)O_2^{\bullet}$ in this range consists of a large peak with FWHM of around 2.5 cm$^{-1}$ sitting on a broad background, whereby the peak makes up roughly half of the absorption. To assure that the decays measured at the peak wavelength of $CH_3C(O)O_2^{\bullet}$ are selective for this radical, kinetics have been measured at different wavelengths. In Figure 1, two decays measured on and off the peak wavelength, obtained following the 248 nm photolysis of $[CH_3C(O)CH_3] = 8.5 \times 10^{15}$ cm$^{-3}$, are shown. No difference in the shape can be observed, and only the overall intensity varies. Therefore, it can be considered that our measurements at 6497.94 cm$^{-1}$ are selective for $CH_3C(O)O_2^{\bullet}$.

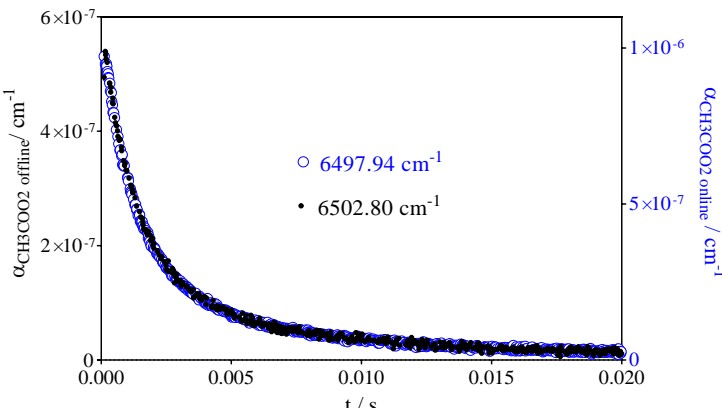

**Figure 1.** Kinetic decays at two different wavelengths, obtained following the 248 nm photolysis of $[CH_3C(O)CH_3] = 8.5 \times 10^{15}$ cm$^{-3}$. Each data point results from one ring-down event, no averaging has been performed.

*2.3. Quantification of HO₂•*

HO$_2^\bullet$ has been detected on the strongest line of the $2\nu_1$ band at 6638.2 cm$^{-1}$. Pressure dependant absorption cross sections in helium [43,44] and in synthetic air [45–47] have been measured several times, but the cross section in pure O$_2$ has been measured only once for a rather small absorption line [48]. Therefore, in the frame of this work, the absorption cross section at 6638.2 cm$^{-1}$ and 100 Torr O$_2$ to $\sigma = 2.0 \times 10^{-19}$ cm$^2$ has been measured using the well-known kinetic method [49,50].

The absorption spectrum of HO$_2^\bullet$ in the near IR is very structured with sharp peaks, thus is it easy to verify the selectivity of the measurement towards HO$_2^\bullet$ by taking decays at the peak wavelength and just next to it, where the HO$_2^\bullet$ absorption is virtually zero. Figure 2 shows an example of both signals (online: black circles, offline: open black circles) measured following the 351 nm photolysis of Cl$_2$ in presence of CH$_3$CHO in 50 Torr helium. The offline HO$_2^\bullet$ signal perfectly matches the CH$_3$C(O)O$_2^\bullet$ signal measured at the same conditions (blue circles, right *y*-axis): it is not unexpected that CH$_3$C(O)O$_2^\bullet$ still absorbs in this wavelength range due to its broad background. From this observation, it can be considered that the HO$_2^\bullet$ concentrations can be obtained in a selective way by taking the difference between online and offline measurements (open squares). The small initial HO$_2^\bullet$ concentration (~$7 \times 10^{11}$ cm$^{-3}$) results from (R9b) and corresponds to ~2% of the CH$_3$C(O)O$_2^\bullet$ concentration (~$3 \times 10^{13}$ cm$^{-3}$) (see Table 1).

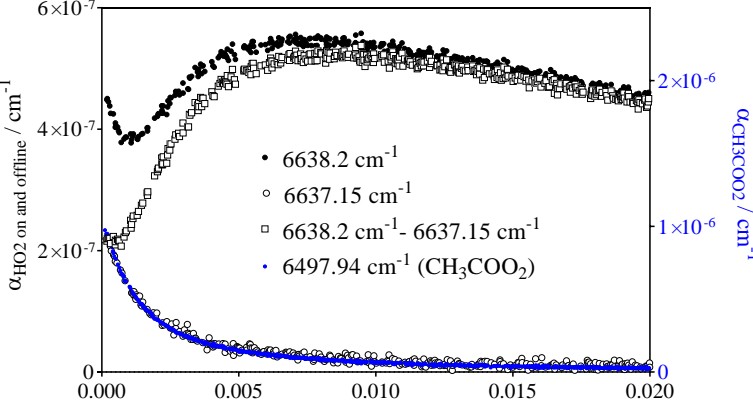

**Figure 2.** Decays obtained following the 351 nm photolysis of $[Cl_2] = 4.5 \times 10^{15}$ cm$^{-3}$ in presence of $[CH_3CHO] = 8.4 \times 10^{14}$ cm$^{-3}$ at a total pressure of 50 Torr helium ($[O_2] = 3 \times 10^{17}$ cm$^{-3}$). Black dots: peak absorption of HO$_2^\bullet$ radicals (6638.2 cm$^{-1}$), open black circles: offline HO$_2^\bullet$ (6637.15 cm$^{-1}$), open squares: the difference between both (left *y*-axis applies). For comparison, the corresponding CH$_3$C(O)O$_2^\bullet$ (blue dots, 6497.94 cm$^{-1}$) has been scaled on the right *y*-axis to match the offline signal.

### 2.4. Quantification of CH3O2•

CH$_3$O$_2$$^{\bullet}$ has been detected on the $\nu_{12}$ transition of the methyl torsion, with the maximum being located at 7488 cm$^{-1}$. The spectrum in this range has been measured several times [50–52] and is, similar to the CH$_3$C(O)O$_2$$^{\bullet}$ spectrum, made of a rather broad background with three distinct peaks. In order to check for the selectivity of the CH$_3$O$_2$$^{\bullet}$ detection, decays were measured at different wavelengths: on top of the three peaks, as well as at different wavelengths on the broad background. Figure 3 summarizes the results.

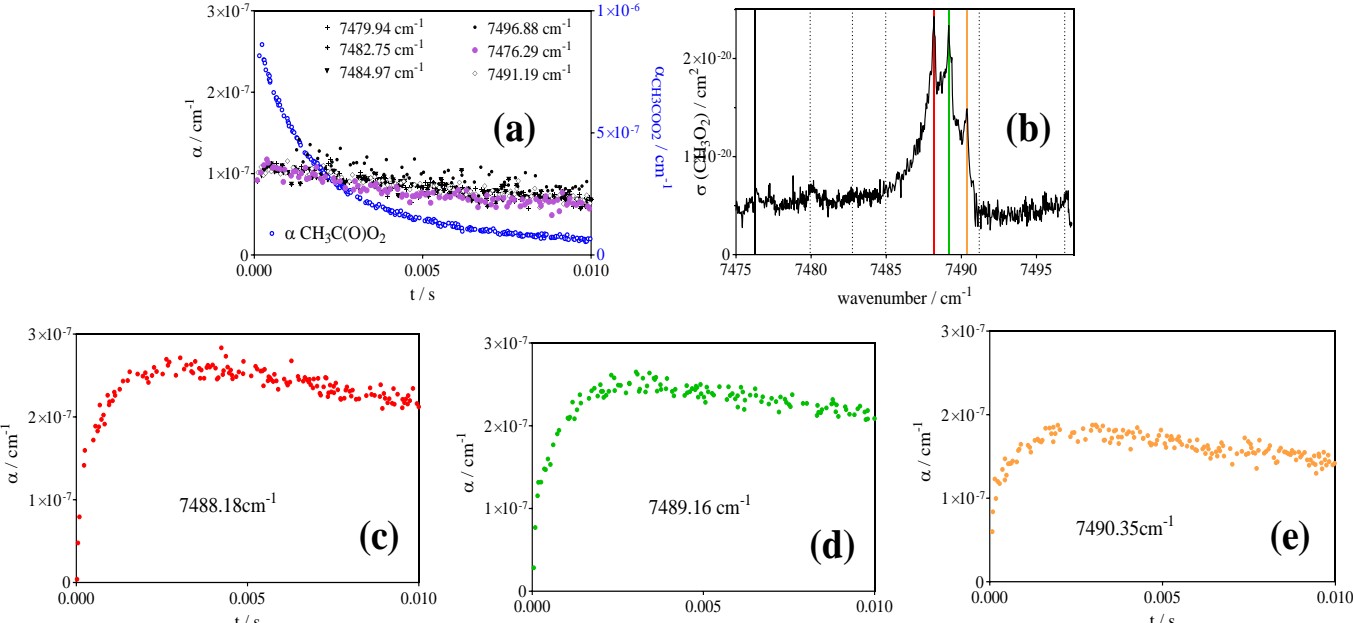

**Figure 3.** Absorption spectrum of CH$_3$O$_2$$^{\bullet}$, adapted from Farago et al. [50] (**b**) and kinetic decays taken following the 351 nm photolysis of [Cl$_2$] = 4.5 × 10$^{15}$ cm$^{-3}$ in presence of [CH$_3$CHO] = 1.66 × 10$^{15}$ at a total pressure of 50 Torr helium ([O$_2$] = 3 × 10$^{17}$ cm$^{-3}$). (**c–e**) show the decays taken at the three peaks of the spectrum, (**a**) shows all decays taken at background wavelengths, indicated by vertical dashed lines in the spectrum. The decay of CH$_3$C(O)O$_2$$^{\bullet}$ is given for information as blue circles in graph a. (right *y*-axis applies).

There is a clear difference in shape between the decays on the peak wavelengths and those on the background wavelengths, an indication that the signal is due to the absorption of at least two different species. The second species should also be transient, i.e., not a stable product from the photolysis, because to make the signal in the background wavelength region look flat, the second absorber must decay roughly on the same time scale as the CH$_3$O$_2$$^{\bullet}$ radical signal increases and should have roughly the same intensity as the absorption due to CH$_3$O$_2$$^{\bullet}$. HO$_2$$^{\bullet}$ is not a possible candidate because (a) it is known that neither the $^{\bullet}$OH vibration overtone [53] nor the $\tilde{A}$-$\tilde{X}$ electronic transition [35,54] of HO$_2$$^{\bullet}$ are located in this wavelength range and (b) one would expect a structured spectrum in the case of HO$_2$$^{\bullet}$. However, it seems that the second absorbing species has a broadband type absorption: all decays in the background region have the same intensity and shape. The second (and major) species present in this system is the CH$_3$C(O)O$_2$$^{\bullet}$ radical. While the spectrum of this radical has been measured in a large wavelength range in the near IR [41], it has never been measured around 7500 cm$^{-1}$, and it is thought to be unlikely that its absorption feature reaches into this range [55]. An initial suspicion concerned the possible formation of the vinoxy peroxy radical, $^{\bullet}$O$_2$CH$_2$CHO, formed from the O$_2$ addition to the initially formed vinoxy radical, $^{\bullet}$CH$_2$CHO [56]. To our knowledge, the $\tilde{A}$-$\tilde{X}$ electronic transition of this radical has never been measured but can be reasonably well expected in this wavelength range because its structure is closer to that of alkyl peroxy radicals than to the acetylperoxy radical. However, the upper limit of the branching fraction for the

formation of the vinoxy radical in the reaction of $Cl^\bullet$-atoms with $CH_3CHO$ is estimated to be only 7% [57], which would demand an unreasonably high absorption cross section, thus making it unlikely that this absorption is due to $^\bullet O_2CH_2CHO$.

In order to remove doubts and to clarify the origin of this absorption, similar experiments have been carried out following the 248 nm photolysis in 100 Torr $O_2$ of acetone. Figure 4 shows a typical example of $[CH_3C(O)CH_3] = 8.5 \times 10^{15}$ cm$^{-3}$ (left graph) and $1.6 \times 10^{16}$ cm$^{-3}$ $Cl_2$ in the presence of $5.9 \times 10^{15}$ cm$^{-3}$ $CH_3CHO$ (right graph). In the first system, only $CH_3O_2^\bullet$ and $CH_3C(O)O_2^\bullet$ (together with low concentrations of $HO_2^\bullet$ and $^\bullet OH$) are expected: the main product ($\approx 70\%$) is $CH_3O_2^\bullet$ (black circles), $CH_3C(O)O_2^\bullet$ (blue circles) is minor, and in the second system $CH_3C(O)O_2^\bullet$ is expected to be the only product. However, in the offline $CH_3O_2^\bullet$ measurements (scaled on the right $y$-axis to match the online $CH_3O_2^\bullet$) of both experiments, a few data points in the first ms show a deviation from the online measurement, strengthening the hypothesis that $CH_3C(O)O_2^\bullet$ is still absorbing around the $CH_3O_2^\bullet$ band. The difference between online and offline measurements is less visible in Figure 4a compared to Figures 3 and 4b for two reasons: (a) in Figure 4a, there is already a high initial $CH_3O_2^\bullet$ concentration, making up the major fraction of the signal, while in Figures 3 and 4b $CH_3O_2^\bullet$ is only formed as a result of the $CH_3C(O)O_2^\bullet$ self-reaction (see below) and (b) in Figure 4a. $CH_3C(O)O_2^\bullet$ is decaying fast through the reaction with excess $CH_3O_2^\bullet$ (see below), thus its impact on the $CH_3O_2^\bullet$ offline signal is decreasing faster. For filled red and black circles in Figure 4b, see paragraph on $Cl^\bullet + CH_3CHO$ as precursor.

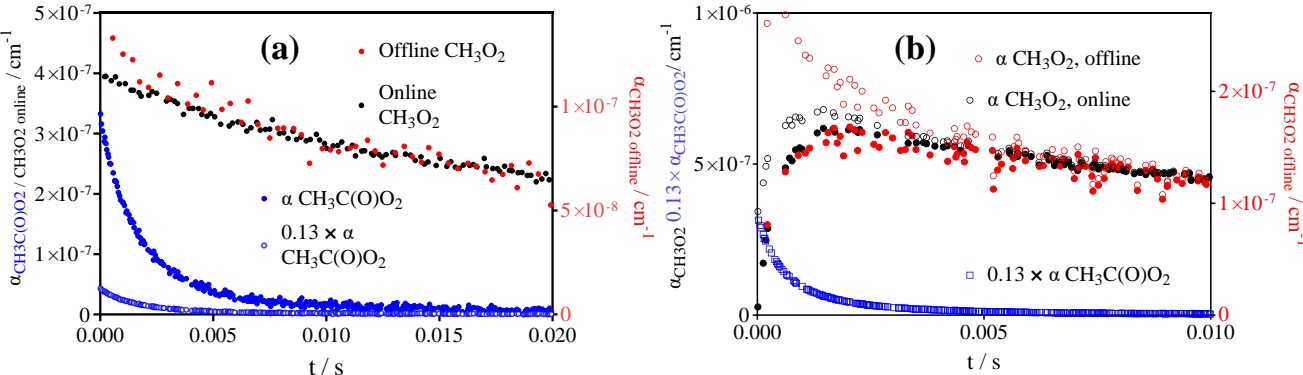

**Figure 4.** (**a**) Kinetic decays following the 248 nm photolysis of $[CH_3C(O)CH_3] = 9.8 \times 10^{15}$ cm$^{-3}$ at 200 Torr. (**b**) Kinetic decays obtained following the 351 nm photolysis of $[Cl_2] = 1.6 \times 10^{16}$ cm$^{-3}$, $[CH_3CHO] = 5.9 \times 10^{15}$ cm$^{-3}$. Black circles show online $CH_3O_2^\bullet$ measurement (left $y$-axis), red circles are offline $CH_3O_2^\bullet$ measurements (right $y$-axis). Blue circles show simultaneous $CH_3C(O)O_2^\bullet$ measurements with open symbols after multiplication with 0.13 to match $CH_3O_2^\bullet$ absorption (see text). Open and filled circles in (**b**) see text on $Cl^\bullet + CH_3C(O)H$ precursor experiments.

Therefore, the signal measured at the online $CH_3O_2^\bullet$ wavelength (7488.18 cm$^{-1}$) is not selective for $CH_3O_2^\bullet$ and must be treated as the sum of two species. Figure 5 shows four absorption signals from Figure 3: the three decays obtained at the peak wavelengths (c-d) as well as one at an offline wavelength (highlighted in magenta in Figure 3a). The $CH_3C(O)O_2^\bullet$ signal is again shown as blue circles. The raw signals are given as open circles, again showing the very different shapes between online and offline signals. The full circles have been obtained by (i) subtracting $0.13 \times \alpha_{CH_3C(O)O_2}$ and (ii) by multiplying the obtained difference with a coefficient appropriate to bring all signals to the same absolute level: 1, 1.02, 1.6, and 4.1 for the red, blue, green, and black circles, in excellent agreement with the $CH_3O_2^\bullet$ absorption cross sections at these wavelengths: 2.4, 2.35, 1.5, and $0.6 \times 10^{-20}$ cm$^2$, respectively.

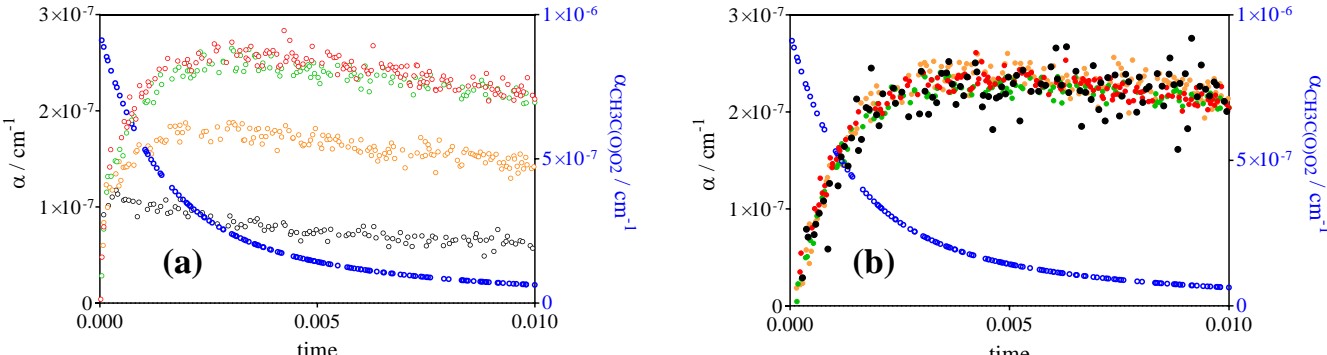

**Figure 5.** Absorption signals from Figure 3 for 4 different wavelengths: the signals at the three peak wavelengths are in colour, an example for the offline (magenta in Figure 3a) is shown in black. (**a**) Raw signals (open symbols), (**b**) signals obtained after subtracting $0.13 \times \alpha_{CH_3C(O)O_2}$ and subsequent multiplication with 1, 1.02, 1.6 and 4.1 for the red, green, orange and black circles, respectively. Blue circles are $\alpha_{CH_3C(O)O_2}$ (right *y*-axis applies).

The factor of $0.13 \times \alpha_{CH_3C(O)O_2}$ has been adjusted so as to give all signals from Figure 3 the same shape after multiplication with the corresponding relative absorption cross section. This test gives confidence that the signals at $7488.18$ cm$^{-1}$ can be used to get selective information on the $CH_3O_2^{\bullet}$ concentration—time profile, as long as the $CH_3C(O)O_2^{\bullet}$ profile can be measured in a selective manner under the same conditions. From these experiments, it can be estimated that the absolute absorption cross section of $CH_3C(O)O_2^{\bullet}$ around $7489$ cm$^{-1}$ is 13% of its value at $6497.94$ cm$^{-1}$, i.e., $\sigma_{CH_3C(O)O_2, 7488\,cm^{-1}} = 4.3 \times 10^{-21}$ cm$^2$.

As a conclusion, our experimental technique allows us to selectively follow the three key radicals playing a role during the study of the title reactions:

- $CH_3C(O)O_2^{\bullet}$ at $6497.94$ cm$^{-1}$ with $\sigma = 3.3 \times 10^{-20}$ cm$^2$
- $HO_2^{\bullet}$ at $6638.2$ cm$^{-1}$ with $\sigma = 2.0 \times 10^{-19}$ cm$^2$ at 100 Torr O$_2$ and $2.72 \times 10^{-19}$ cm$^2$ at 50 Torr He after subtracting the offline signal measured at $6637.15$ cm$^{-1}$
- $CH_3O_2^{\bullet}$ at $7489.16$ cm$^{-1}$ with $\sigma = 2.4 \times 10^{-20}$ cm$^2$ after subtracting $0.13 \times \alpha_{CH_3C(O)O_2}$ such as measured at $6497.94$ cm$^{-1}$ (or, for convenience, by representing $[CH_3O_2^{\bullet}]$ as $[CH_3O_2^{\bullet}] + 0.179 \times [CH_3C(O)O_2^{\bullet}]$, with $0.179 = 4.3 \times 10^{-21}$ cm$^2$/$2.4 \times 10^{-20}$ cm$^2$).

$^{\bullet}$OH radicals, which are formed in the reaction of $CH_3CO^{\bullet} + O_2$ as well as being produced by the cross reaction between $CH_3C(O)O_2^{\bullet} + HO_2^{\bullet}$, can in principle be quantified by cw-CRDS in the near IR, and absorption cross sections have been determined [58]. Some tests did allow the detecting of $^{\bullet}$OH following the photolysis of $CH_3C(O)C(O)CH_3$ and $CH_3C(O)CH_3$ and resulting from (R9b), but the S/N ratio was poor: the $^{\bullet}$OH lifetime is short in the presence of the precursors and peroxy radicals [2], and thus the concentrations are low. Moreover, $CH_3C(O)O_2^{\bullet}$ is absorbing in this wavelength range as well as another species (a stable reaction product, possibly $CH_3OOH$), so a rather small, short-lived $^{\bullet}$OH-signal would be needed to be extracted from online-offline measurements. Therefore, no attempts were made to include $^{\bullet}$OH signals in the modelling procedure. However, the initial $^{\bullet}$OH concentration could be estimated and was roughly the same as the initial $HO_2^{\bullet}$ concentration under the same conditions. For this reason, a simplified reaction (R9b) leading to equal concentrations of $^{\bullet}$OH and $HO_2^{\bullet}$ (see Table 3) has been used during modelling. The concentration was always small compared to $CH_3O_2^{\bullet}$ or $CH_3C(O)O_2^{\bullet}$, and hence the influence of (R9b) on the retrieved results for (R1) and (R2) is negligible. LIF measurements, also available in our set-up [37,59], were not possible due to strong quenching of the fluorescence, because experiments have been performed mostly in 100 Torr O$_2$ in order to rapidly convert $CH_3O^{\bullet}$ radicals into $HO_2^{\bullet}$.

**Table 3.** Mechanism used for fitting the different profiles under all conditions.

| No | Reaction | $k/10^{-11}$ cm$^3$s$^{-1}$ | References |
|---|---|---|---|
| 1 | $2\ CH_3C(O)O_2{}^\bullet \rightarrow \rightarrow \rightarrow 2\ CH_3O_2{}^\bullet$ | $1.3 \pm 0.3$ * | This work |
| 2a | $CH_3C(O)O_2{}^\bullet + CH_3O_2{}^\bullet \rightarrow CH_3O^\bullet +$ $CH_3C(O)O^\bullet + O_2$ | $1.35 \pm 0.2$ * | This work |
| 2b | $\rightarrow$ molecular products | $0.65 \pm 0.2$ * | This work |
| 3a/b | $CH_3C(O)O_2{}^\bullet + HO_2{}^\bullet \rightarrow$ molecular products | 0.86 | [27] |
| 3c | $\rightarrow CH_3C(O)O^\bullet + {}^\bullet OH + O_2$ | 0.86 | |
| 5 | ${}^\bullet CH_3 + O_2\ (+M) \rightarrow CH_3O_2{}^\bullet\ (+M)$ | 0.035 | [60] |
| 6 | $CH_3O^\bullet + O_2 \rightarrow CH_2O + HO_2{}^\bullet$ | $1.9 \times 10^{-4}$ | [30] |
| 9a | $CH_3CO^\bullet + O_2\ (+ M) \rightarrow CH_3C(O)O_2{}^\bullet\ (+ M)$ | 0.7 | [30] |
| 9b | $CH_3CO^\bullet + O_2 \rightarrow {}^\bullet CH_2CO +$ $HO_2{}^\bullet/CH_2C(O)O^\bullet + {}^\bullet OH$ | Varied | see Table 2 |
| 10 | $CH_3CO^\bullet \rightarrow {}^\bullet CH_3 + CO$ | Varied | see Table 2 |
| 11 | $2\ HO_2{}^\bullet \rightarrow H_2O_2 + O_2$ | 0.17 | [61] |
| 12a | $2\ CH_3O_2{}^\bullet \rightarrow 2\ CH_3O^\bullet + O_2$ | 0.013 | [30] |
| 12b | $\rightarrow CH_3OH + CH_2O + O_2$ | 0.022 | |
| 13 | $CH_3O_2{}^\bullet + HO_2{}^\bullet \rightarrow CH_3OOH$ | 0.52 | [30] |
| 14 | ${}^\bullet OH + CH_3CHO \rightarrow CH_3CO^\bullet + H_2O$ | 1.6 | [30] |
| 15 | ${}^\bullet OH + CH_3C(O)CH_3 \rightarrow CH_2C(O)CH_3 + H_2O$ | 0.022 | [30] |
| 16 | ${}^\bullet OH + CH_3C(O)C(O)CH_3 \rightarrow$ $CH_2C(O)C(O)CH_3 + H_2O$ | 0.025 | [62] |
| 17 | ${}^\bullet OH + CH_3O_2{}^\bullet \rightarrow CH_3O^\bullet + HO_2{}^\bullet$ | 12 | [3] |
| 18 | ${}^\bullet OH + CH_3C(O)O_2{}^\bullet \rightarrow CH_3C(O)O_3H$ | 10 | [12,63] |
| 19 | $Cl^\bullet + CH_3CHO \rightarrow CH_3CO^\bullet + HCl$ | 7.2 | [30] |
| 20 | $Cl_2 + CH_3CO^\bullet \rightarrow CH_3COCl + Cl$ | 4.3 | [64] |
| 21 | $HO_2{}^\bullet, CH_3O_2{}^\bullet, CH_3C(O)O_2{}^\bullet \rightarrow$ diffusion | $1\text{–}2\ \text{s}^{-1}$ | |

* Uncertainties are only based on unacceptable deviations of the model from measurements using the absorption cross sections such as given in the text.

## 3. Results and Discussion

### 3.1. Photolysis of CH$_3$C(O)CH$_3$ and CH$_3$C(O)C(O)CH$_3$

Typical concentration–time profiles obtained following the 248 nm photolysis of three different concentrations of acetone (left) and biacetyl (right) are presented in Figure 6a,b showing [CH$_3$C(O)O$_2{}^\bullet$] and Figure 6c,d showing [CH$_3$O$_2{}^\bullet$], i.e., $\alpha$(CH$_3$O$_2{}^\bullet$) converted with $\sigma = 2.4 \times 10^{-20}$ cm$^2$, thus representing the sum of [CH$_3$O$_2{}^\bullet$] + 0.179 × [CH$_3$C(O)O$_2{}^\bullet$]. Figure 6e,f show [HO$_2{}^\bullet$]. The full lines represent the model given in Table 3, whereby the dashed line for the CH$_3$O$_2{}^\bullet$ profiles represent the above sum and the full line represents the modelled [CH$_3$O$_2{}^\bullet$] profile. The model in Table 3 contains only rate constants from the literature, except for the two title reactions: the self-reaction of CH$_3$C(O)O$_2{}^\bullet$ (R1) and its cross reaction with CH$_3$O$_2{}^\bullet$ (R2). It turned out that the profiles were not sensitive to the rate constant of the reaction with HO$_2{}^\bullet$ radicals (R3), and therefore the result from the most recent measurements [27], together with a branching ratio of 0.5 for the radical channel, has been used.

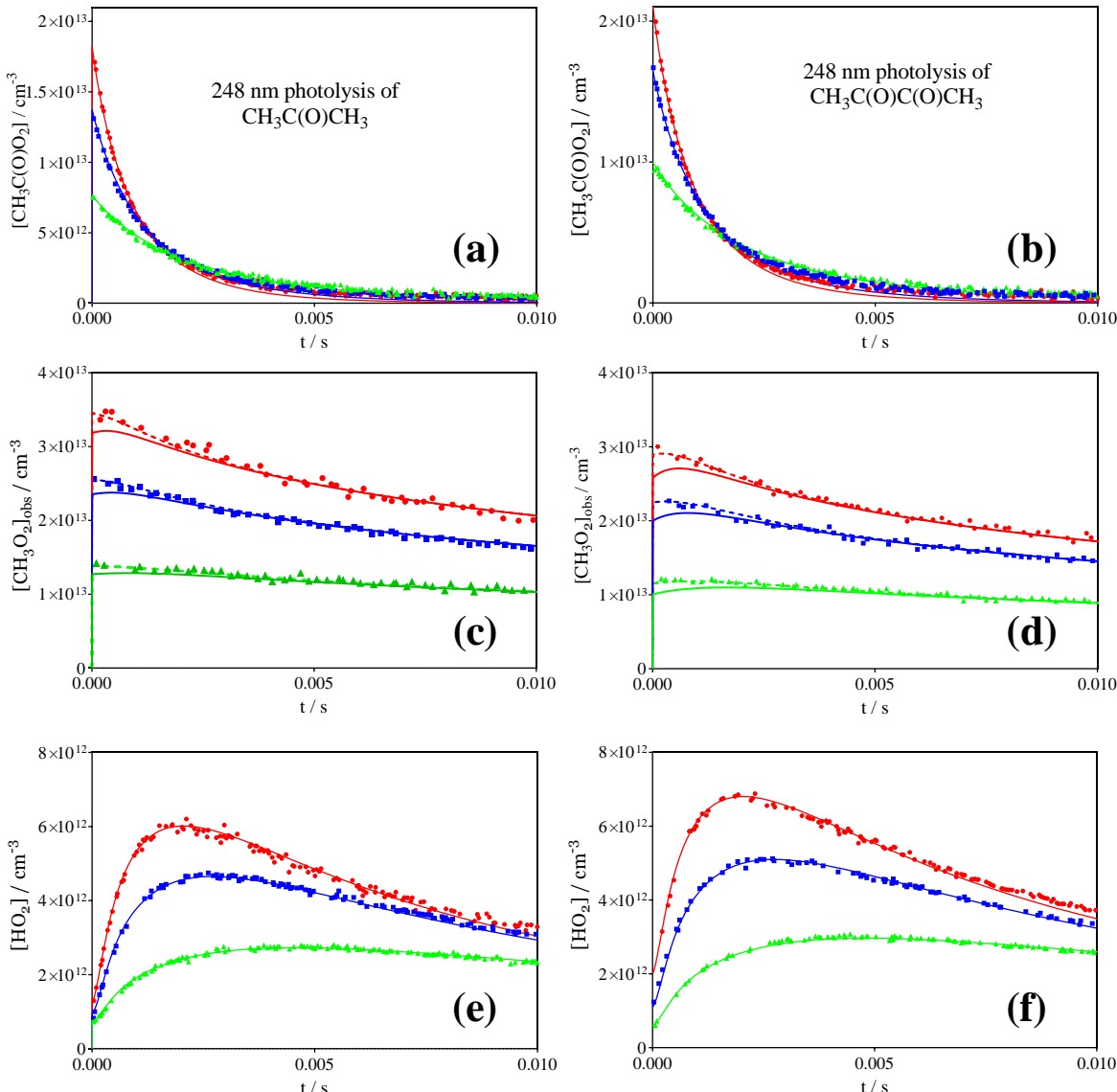

**Figure 6.** Concentration-time profiles of $CH_3C(O)O_2^\bullet$ (**a,b**), $CH_3O_2^\bullet$ (**c,d**) et $HO_2^\bullet$ (**e,f**) measured simultaneously following the 248 nm photolysis of 3 different concentrations of $CH_3C(O)CH_3$ [$CH_3C(O)CH_3$] = 1.21 (red), 0.86 (blue) and $0.4\times10^{16}$ cm$^{-3}$ (green) (left raw) and $CH_3C(O)C(O)CH_3$ [$CH_3C(O)C(O)CH_3$] = 6 (red), 2.56 (blue) and $0.93\times10^{16}$ cm$^{-3}$ (green) (right raw) in 100 Torr $O_2$. Full lines: simulation following the model and rate constants as given in Table 3. For the [$CH_3O_2^\bullet$] profiles, the dashed lines show [$CH_3O_2^\bullet$] + 0.179 × [$CH_3C(O)O_2^\bullet$]. Experimental $HO_2^\bullet$ profiles have been decreased by around 10% for the lowest radical concentration, i.e., the model systematically underestimates $HO_2^\bullet$ at low overall radical concentrations.

It can be seen that the radical profiles for all three species are very well reproduced for all conditions and precursors. However, some small deviations have been systematically observed, and cannot currently be explained:

The modelled $CH_3C(O)O_2^\bullet$ profile is very well reproduced over the first few ms (up to 70–80% decay of its initial concentration), but decays become too fast thereafter, and are especially visible for the highest radical concentrations. This behaviour could not be corrected by adapting rate constants or branching ratios, because at longer reaction times the decay of $CH_3C(O)O_2^\bullet$ is nearly exclusively governed by reaction with $CH_3O_2^\bullet$, which itself is very well reproduced: slowing down the rate constant of (R2) does not remediate, because it would also have a similar impact at short reaction times, and would thus make the $CH_3C(O)O_2^\bullet$ decay too slow in a short time. Trying to remediate by increasing $k_1$

does not bring success neither, because (a) it increases the decay rate even more at short reaction times, when $[CH_3C(O)O_2^\bullet]$ is still high, and (b) it will increase the $CH_3O_2^\bullet$ concentration well above the measurements. An explanation for this could be that, if a reaction product would absorb at the same wavelength than $CH_3C(O)O_2^\bullet$, it is thought that the $CH_3C(O)O_2^\bullet$ measurement is selective, because the profiles "online $CH_3C(O)O_2^\bullet$ " and "offline $HO_2^\bullet$" always have the same shape, despite a difference in the absorption cross section for $CH_3C(O)O_2^\bullet$ of a factor of four between both wavelengths (see Figure 2). Another explanation could be an unidentified continuous formation of $CH_3C(O)O_2^\bullet$. However, this is unlikely as it was observed with all precursors and a continuous formation would result in an increased formation of $CH_3O_2^\bullet$ due to a sustained self-reaction. Currently, this deviation cannot be explained, but as it occurs only when $[CH_3C(O)O_2^\bullet]$ is already low, it is highly probable that the impact on the retrieved rate constants for (R1) and (R2) is very minor.

Another small deviation observed for all precursors and all conditions was a slight, systematic underestimation of $HO_2^\bullet$ at low radical concentrations: the experimental $[HO_2^\bullet]$ profiles in a typical series such as shown in Figure 6 need to be decreased by around 10% for the lowest concentration. Alternatively, if $HO_2^\bullet$ was tentatively increased through either decreasing rate constants for its consumptions or increasing yields for its production, it was overestimated by around 10% at the highest radical concentrations. All $HO_2^\bullet$ in the mechanism of Table 3 originates from radical–radical reactions, and thus it will not be possible to bring into agreement these profiles under all conditions. It seems that a small fraction of $HO_2^\bullet$ radicals originates from a unimolecular (or pseudo-first order) process. It could also be the case that a sizeable fraction of $HO_2^\bullet$ is complexed with the precursor even at room temperature, as proposed by Hui et al. [27] However, no parameter could be found that led to satisfactory results over the entire concentration range. Moreover, the effect is very similar for acetone and biacetyle, which would suppose a similar equilibrium constant.

Interestingly, experiments at a total pressure of 200 Torr $O_2$ can be very well simulated for all radicals with the rate constants from Table 3, except that the profile for $HO_2^\bullet$ radicals decays too fast at longer reaction times. Figure 7 shows a series of measurements following the 248 nm biacetyle photolysis at a total pressure of 200 Torr. $CH_3O_2^\bullet$ and $CH_3C(O)O_2^\bullet$ profiles are very well reproduced, but experimental $HO_2^\bullet$ profiles decay much faster than predicted by the model (full lines). An increase of the rate constant for the $HO_2^\bullet$ self-reaction from $1.7 \times 10^{-12}$ $cm^3s^{-1}$ to $5 \times 10^{-12}$ $cm^3s^{-1}$ leads to good reproduction of the $HO_2^\bullet$ profiles (dashed black line in $HO_2^\bullet$ profiles, Figure 7c). While it is known that this rate constant is pressure dependent, only a small increase up to around $2 \times 10^{-12}$ $cm^3s^{-1}$ would be expected with an increase in $O_2$ pressure from 100 to 200 Torr [61]. The effect was on the same order when acetone photolysis was the precursor (the rate constant for the $HO_2^\bullet$ self-reaction had to be increased to $4 \times 10^{-12}$ $cm^3s^{-1}$ to well-reproduce the decay of the $HO_2^\bullet$ profiles). Such an observation points towards a strong chaperone effect of both precursors, as already proposed by Hui et al., and more experiments focussed on this subject are planned in the future. On the other hand, $HO_2^\bullet$ decays during 100 Torr experiments with comparable precursor concentration can be well reproduced using $1.7 \times 10^{-12}$ $cm^3s^{-1}$ for the $HO_2^\bullet$ self-reaction. The increase of this rate constant has a negligible impact on the profiles of the other two species and on the retrieved results for (R1) and (R2): this is expected, as the cross reactions with $HO_2^\bullet$ are only minor paths for both species under our conditions, where the $HO_2^\bullet$ concentration is 4–5 times lower than the concentration of the two other species. Therefore, the influence of this observation is at the most very minor with respect to the retrieved rate constants and branching ratios of the title reactions. Experiments at higher temperature are planned in the future to exclude any influence of complexation on the $HO_2^\bullet$ profile.

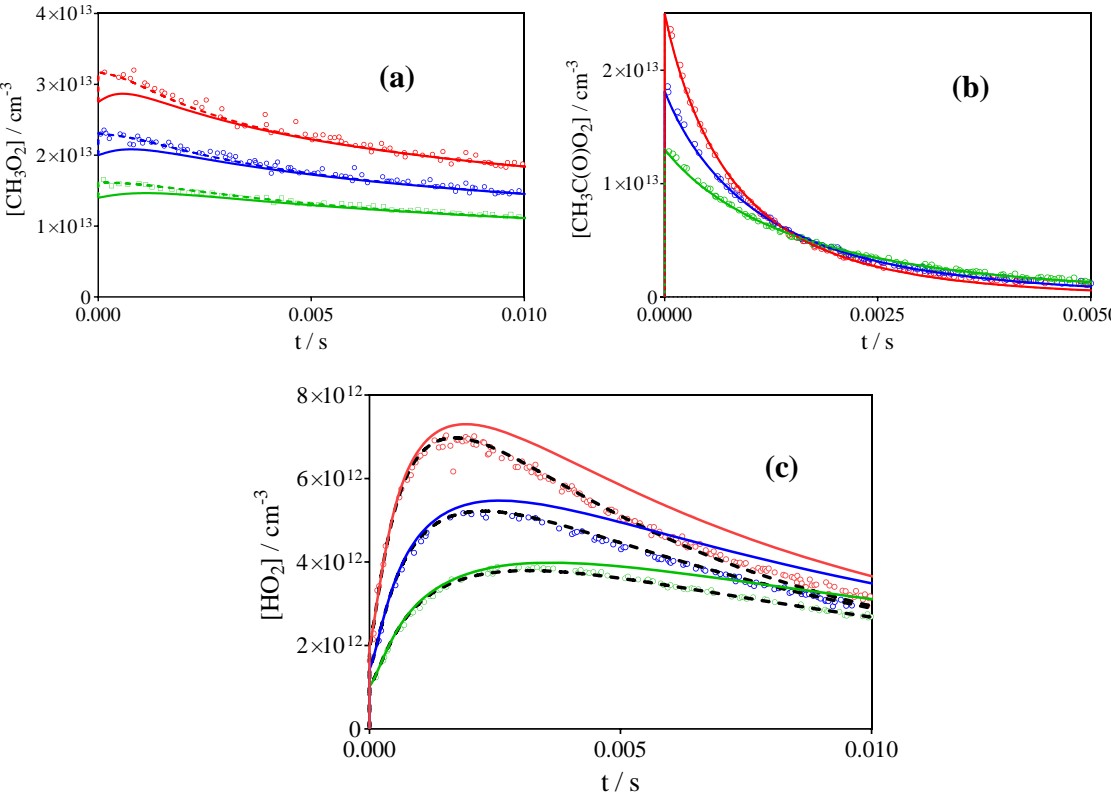

**Figure 7.** Concentration-time profiles of $CH_3O_2^\bullet$ (**a**), $CH_3C(O)O_2^\bullet$ (**b**) and $HO_2^\bullet$ (**c**) following the 248 nm photolysis of 3 different concentration of $CH_3C(O)C(O)CH_3$ (8.67 (red), 5.79 (blue) and $2.85 \times 10^{16}$ cm$^{-3}$ (green)) at a total pressure of 200 Torr $O_2$. Full lines simulation with rate constants from Table 3, dashed black lines using a rate constant for $HO_2^\bullet$ self-reaction of $5 \times 10^{-12}$ cm$^3$ s$^{-1}$. Changes in $CH_3O_2^\bullet$ and $CH_3C(O)O_2^\bullet$ profiles are not visible and have been omitted from the graph.

### 3.2. Comparison with Literature Rate Constants

Figure 8 shows all three radical profiles ($CH_3O_2^\bullet$ in blue, $CH_3C(O)O_2^\bullet$ in red and $HO_2^\bullet$ in green) for the highest initial radical concentration of the acetone photolysis experiment at 100 Torr from Figure 6 (red symbols, left column). The four graphs show different models using the mechanism from Table 3 with rate constants for (R1) and (R2) from Table 1 (the model of Roehl et al. [16] is close to IUPAC and is not reproduced here): Figure 8a represents the best fit as deduced in this work and as given in Table 3. Figure 8b–d present the same model except for the rate constants for (R1) and (R2) that have been changed to different literature results. Figure 8b represents the model such as proposed by Maricq and Szente [17]: even though the rate constant for the self-reaction of $CH_3C(O)O_2^\bullet$ is faster than in our model, the $CH_3C(O)O_2^\bullet$ profile decays not fast enough compared to the observation. This is due to the fact that the authors propose only molecular products for the cross reaction with $CH_3O_2^\bullet$—the $HO_2^\bullet$ profile is not reproduced at all because the radical path of (R2) is the major source of $HO_2^\bullet$—, and thus the removal of $CH_3C(O)O_2^\bullet$ by both cross reactions is too slow. The two other models have higher radical yields for the cross reaction with $CH_3O_2^\bullet$ (0.9 for IUPAC [30] and 0.65 for Villenave and Lesclaux [28]), but around a factor of two lower rate constant for $k_2$. This leads to more or less acceptable $HO_2^\bullet$ profiles, however the $CH_3O_2^\bullet$ profiles are not well reproduced: in both cases, the concentration initially increases before decaying roughly at the observed rate. This initial increase in $CH_3O_2^\bullet$ concentrations has never been observed in our experiments. Moreover, the lower rate constant for (R2) leads to decays for $CH_3C(O)O_2^\bullet$ that are much slower than the observed profiles. A comparison with the predictions of the semi-empirical study [29] is not shown, because no branching ratios are predicted, which is indispensable for the prediction of concentration–time profiles. However, the rate constant for (R2) predicted in

the semi-empirical study, based on the stabilization energy of the tetroxide intermediate, is $7 \times 10^{-12}$ cm$^3$s$^{-1}$ even below the lowest experimental value and it can therefore be supposed that the semi-empirical method is not reliable for this type of cross-reaction. To our knowledge, no theoretical calculations concerning mechanism and rate constants of (R1) and (R2), and more importantly the branching ratio for (R2), have been carried out, but that would certainly be interesting.

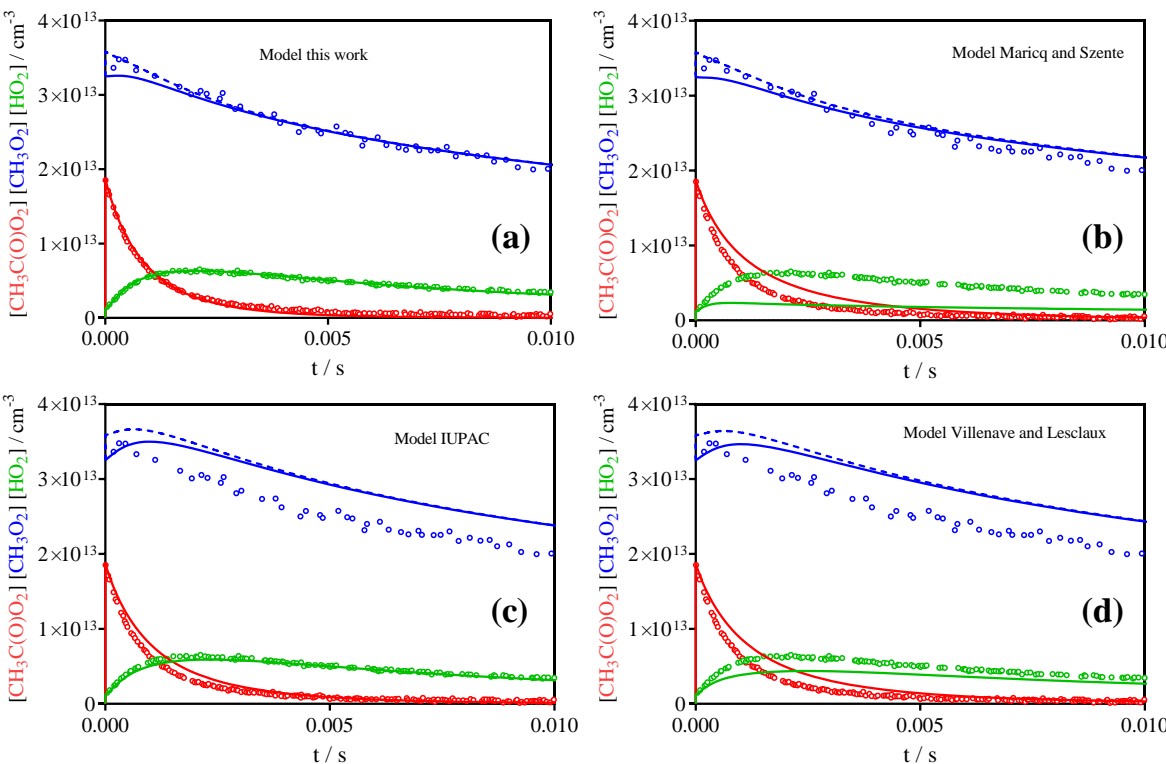

**Figure 8.** Highest radical concentration from 248 nm photolysis of CH$_3$C(O)CH$_3$ shown in Figure 6 (red symbols from the left column): CH$_3$O$_2{}^\bullet$ profile in blue, HO$_2{}^\bullet$ profile in green and CH$_3$C(O)O$_2{}^\bullet$ profile in red. (**a**) model as given in Table 3, (**b**) same model, but rate constants for (R1) and (R2) such as propose by Maricq and Szente [17], (**c**) same model, but $k_1$ and $k_2$ such as recommended currently by IUPAC [30], (**d**) same, but $k_1$ and $k_2$ from Villenave and Lesclaux [28] (see Table 1).

### 3.3. Reaction of Cl-Atoms with CH$_3$CHO as Precursor

In all earlier studies summarized in Table 1, CH$_3$C(O)O$_2{}^\bullet$ radicals have been prepared by H-atom abstraction from CH$_3$CHO through reaction with Cl-atoms, whereby Cl$^\bullet$-atoms have been generated from 351 nm photolysis of Cl$_2$, a wavelength where CH$_3$CHO does not absorb. The present study is the first one that used different precursors. Therefore, this precursor was also tested in this experiment. Another reason to try this precursor was that this reaction system should be a clean source of CH$_3$C(O)O$_2{}^\bullet$ radicals next to low concentrations of O$_2$CH$_2$C(O)H, $^\bullet$OH, and HO$_2{}^\bullet$ but no initial CH$_3$O$_2{}^\bullet$. This reaction system should be especially suited for measuring the rate constant of (R1), as the initial decay of CH$_3$C(O)O$_2{}^\bullet$ radicals is not perturbed by already present CH$_3$O$_2{}^\bullet$ radicals. Figure 9 shows the profiles of the three radical species obtained following the 351 nm photolysis of [Cl$_2$] = (0.34–1.6) $\times$ 10$^{16}$ cm$^{-3}$ in the presence of [CH$_3$CHO] = 5.9 $\times$ 10$^{15}$ cm$^{-3}$, together with the simulation using the model from Table 3. To our great surprise, the measured CH$_3$O$_2{}^\bullet$ profiles were not at all reproduced by the model, as the predicted CH$_3$O$_2{}^\bullet$ concentration rises much too fast and too high. Moreover, the predicted HO$_2{}^\bullet$ rises too fast and too high, which is a direct consequence of the high CH$_3$O$_2{}^\bullet$ concentration: HO$_2{}^\bullet$ is nearly exclusively formed as a product of (R2). CH$_3$C(O)O$_2{}^\bullet$ also decays too fast, which is also a consequence of the too high CH$_3$O$_2{}^\bullet$ concentration: the dashed magenta line in

Figure 9 presents the fraction of $CH_3C(O)O_2^{\bullet}$ radicals that has been removed through reaction with $CH_3O_2^{\bullet}$ for the highest radical concentration (blue symbols). Decreasing the rate constant for (R1) to $k_1 = 1 \times 10^{-11}$ cm$^3$s$^{-1}$ improves the agreement for $CH_3C(O)O_2^{\bullet}$ profiles somewhat (although still too fast after around 70% of the decay, as for the other precursors), but even then the two other radicals are still much overestimated.

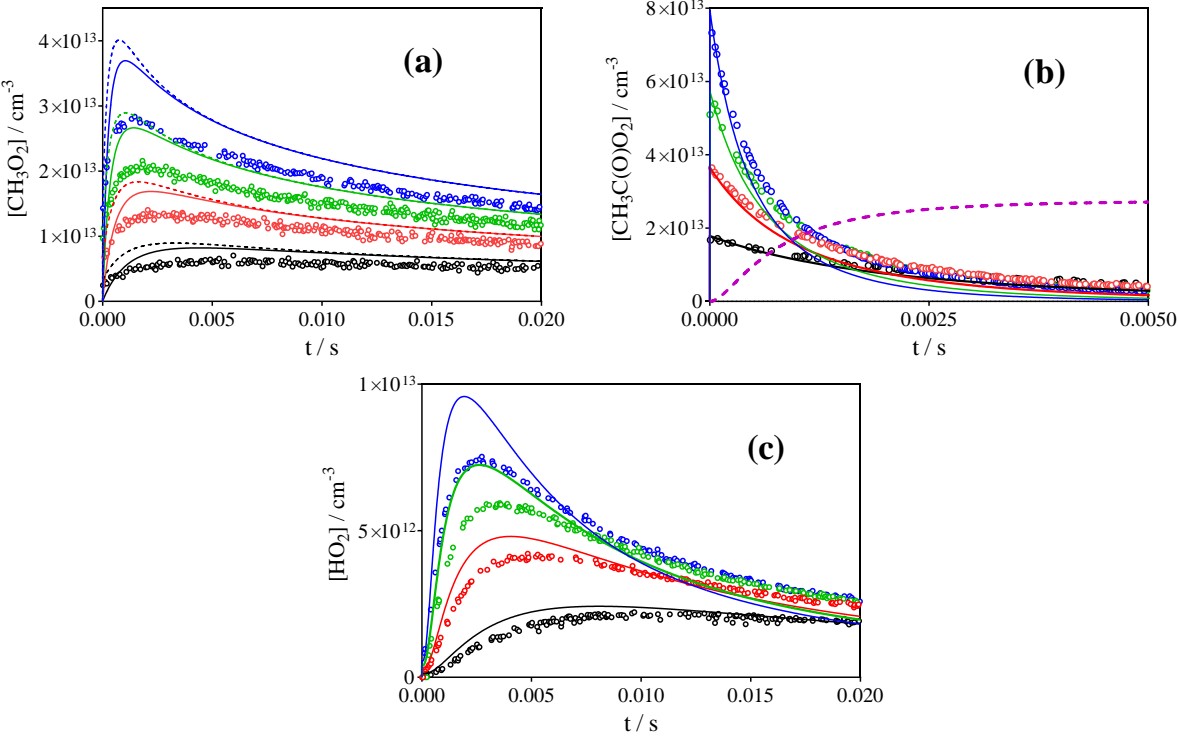

**Figure 9.** Concentration-time profiles for $CH_3O_2$ (**a**), $CH_3C(O)O_2$ (**b**) and $HO_2$ (**c**), obtained following the photolysis of $[Cl_2] = = 1.6$ (blue), 1.17 (green), 0.77 (red) and $0.34 \times 10^{16}$ cm$^{-3}$ (black) in presence of $[CH_3CHO] = 5.9 \times 10^{15}$ cm$^{-3}$ at 100 Torr $O_2$. Full lines present the model from Table 3, dashed line in $CH_3O_2^{\bullet}$ profiles represent $[CH_3O_2^{\bullet}] + 0.179 \times [CH_3C(O)O_2^{\bullet}]$.

To test for unidentified secondary chemistry of $Cl^{\bullet}$-atoms or acetaldehyde, experiments have been carried out using different ratios of $Cl^{\bullet}/CH_3CHO$: Figure 10 shows the three radical profiles of experiments using identical $Cl_2$ concentrations and photolysis energies, but the concentration of $CH_3CHO$ has been changed by a factor of 3, from $(3.1 \text{ to } 9.2) \times 10^{15}$ cm$^{-3}$. No change in any of the three radical profiles is observed, from which it can be concluded that $CH_3CHO$ is not involved in the reaction system other than its reaction with $Cl^{\bullet}$-atoms. Similar experiments have been carried out by changing the repetition rate of the photolysis laser between 1 and 0.1 Hz (experiments were typically carried out at 0.3 Hz) to test for possibly remaining reaction products. However, no change is observed, other than a slight increase of $HO_2^{\bullet}$ with increasing repetition rate (possibly due to reaction of $Cl^{\bullet}$-atoms with remaining reaction products such as $CH_2O$) (Figure 10d). $CH_3CO^{\bullet}$ radicals are probably less energetic when using this precursor (H-abstraction from $CH_3CHO$) compared to the photolysis of acetone or biacetyle. To test if the difference in internal energy of the initial radical can bias the results, $Cl^{\bullet} + CH_3CHO$ experiments in 50 Torr helium have been carried out, but there was no difference in the profiles of the different radicals compared to the 100 Torr $O_2$ experiments: there was always too much $CH_3O_2^{\bullet}$.

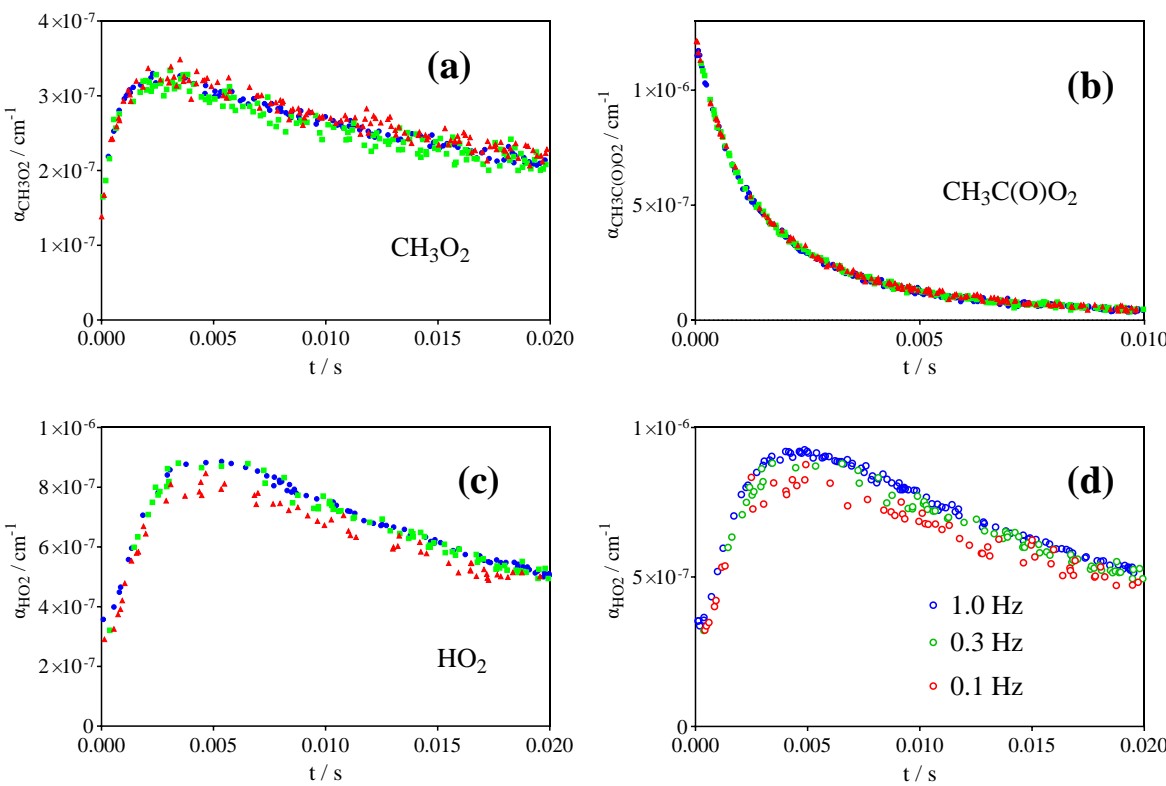

**Figure 10.** Test for unidentified secondary chemistry involving CH$_3$CHO or reaction products remaining within the photolysis volume: (**a**) CH$_3$O$_2$$^\bullet$, (**b**) CH$_3$C(O)O$_2$$^\bullet$, (**c**) HO$_2$$^\bullet$ profiles at different CH$_3$CHO concentrations (Cl$_2$ = 7.7 × 10$^{15}$ cm$^{-3}$, CH$_3$CHO = 3.1 (blue), 6.2 (green) and 9.2× 10$^{15}$ cm$^{-3}$ (red)). (**d**) HO$_2$$^\bullet$ profiles at different photolysis repetition rates.

From the above discussions, it can be concluded with reasonable confidence that the CH$_3$O$_2$$^\bullet$ measurements are selective for CH$_3$O$_2$$^\bullet$ in the same way that they are for the other precursors: Figure 4b shows CH$_3$O$_2$$^\bullet$ online and offline measurements and the simultaneously obtained CH$_3$C(O)O$_2$$^\bullet$ absorption profile, multiplied by 0.13. The absorption at t = 0 s starts at the same point, indicating that all absorption at t = 0 s is due to CH$_3$C(O)O$_2$$^\bullet$ and no CH$_3$O$_2$$^\bullet$ is formed initially, as expected. By treating the signal as shown in Figure 5, i.e., subtracting the absorbance due to CH$_3$C(O)O$_2$$^\bullet$ from both online (black open circles) and offline (red open circles), identical signals (filled black and red circles) are obtained, indicating that with this precursor the CH$_3$O$_2$$^\bullet$ signal can also be taken as selective if treated as the sum of $\alpha$(CH$_3$O$_2$$^\bullet$) + 0.13 × $\alpha$(CH$_3$C(O)O$_2$$^\bullet$).

Finally, the rate constants from the literature, which all used Cl$^\bullet$ + CH$_3$CHO as precursor, have been tested against the profiles of the three radicals, and the result of the corresponding models is shown in Figure 11 for the highest radical concentration from Figure 9 (CH$_3$O$_2$$^\bullet$ in blue, CH$_3$C(O)O$_2$$^\bullet$ in red and HO$_2$$^\bullet$ in green): Figure 9a shows again the model from Table 3. Figure 9b represents $k_1$ and $k_2$ as given by Maricq and Szente [17]: the CH$_3$C(O)O$_2$$^\bullet$ decay is reasonably well reproduced, but this is again the result of the branching ratio of (R2), which predicts no radical products, leading to much slower cross reactions for CH$_3$C(O)O$_2$$^\bullet$. However, again, the HO$_2$$^\bullet$ profile is not at all reproduced, and even with this model the CH$_3$O$_2$$^\bullet$ concentration is still strongly overpredicted. The situation is similar for the two other models: none of them can reasonably well reproduce all three radical profiles.

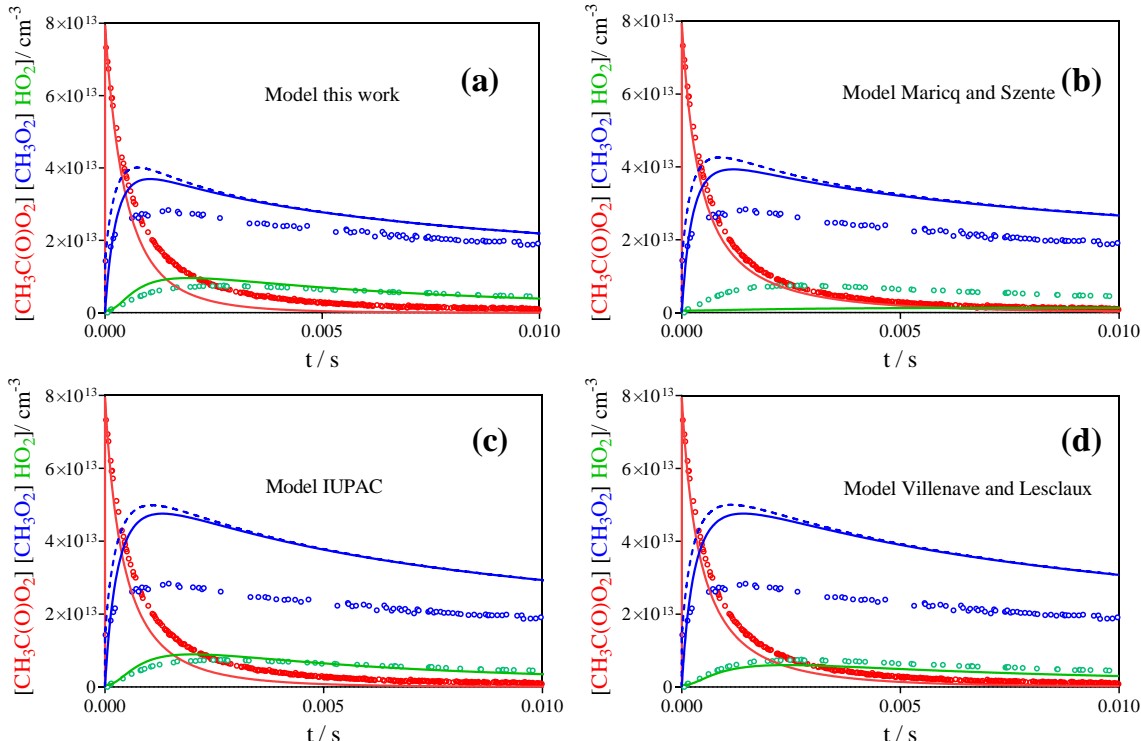

**Figure 11.** Highest radical concentration from Figure 9: $[Cl_2] = 1.6 \times 10^{16}$ cm$^{-3}$ (CH$_3$O$_2$$^\bullet$ in blue, CH$_3$C(O)O$_2$$^\bullet$ in red and HO$_2$$^\bullet$ in green) with different models, such as given in Table 1. Dashed blue lines represent $[CH_3O_2{}^\bullet] + 0.179 \times [CH_3C(O)O_2{}^\bullet]$, which should agree with measurements.

Therefore, no explanation can currently be given for the strong difference in radical profiles between the different precursors, so the mystery remains. Some unidentified secondary chemistry involving Cl$^\bullet$, Cl$_2$, or HCl might take place.

## 4. Conclusions

The rate constant of the self-reaction of CH$_3$C(O)O$_2$$^\bullet$ radicals as well as the rate constant and branching ratio for the radical path for its cross reaction with CH$_3$O$_2$$^\bullet$ radicals has been measured by following the concentration time profiles of the three key radicals CH$_3$C(O)O$_2$$^\bullet$, CH$_3$O$_2$$^\bullet$ and HO$_2$$^\bullet$ in a selective way. The rate constant of the self-reaction has been found as $k_1 = (1.35 \pm 0.3) \times 10^{-11}$ cm$^3$s$^{-1}$, in good agreement with current recommendations. However, the rate constant for the cross reaction has been found as $k_2 = (2.0 \pm 0.4) \times 10^{-11}$ cm$^3$s$^{-1}$, which around two times faster than currently recommended. The yield for the radical maintaining pathway (R2a) has been found as $\alpha = 0.67$, which is slightly below the current IUPAC recommendation (0.9). Some systematic, unexplained deviations between model and measurement persist: the CH$_3$C(O)O$_2$$^\bullet$ concentration seems to be maintained at long reaction times at a higher concentration than predicted by the model. This could be explained by the reaction product absorbing at the same wavelength as CH$_3$C(O)O$_2$$^\bullet$, even though tests have been carried out which do not confirm this hypothesis. Another unexplained deviation persists in that at higher total pressures (200 Torr O$_2$ instead of 100 Torr O$_2$), the HO$_2$$^\bullet$ concentration decays faster than the model predicts: this could be due to a complexation of HO$_2$$^\bullet$ with the precursor and a resulting increased rate constant for the self-reaction. However, this explanation, even though already mentioned by Hui et al. [27], is not satisfying, as the profiles can be very well reproduced using the same precursor concentrations in 100 Torr O$_2$ without accounting for complexation. The most mysterious, unexplained observation occurred when the reaction of Cl$^\bullet$-atoms with CH$_3$CHO was the precursor for CH$_3$C(O)O$_2$$^\bullet$ radicals: the observed CH$_3$O$_2$$^\bullet$ concentration was much higher than predicted by the model. Unidentified secondary chemistry involving Cl$^\bullet$, Cl$_2$, or HCl might be involved, but currently no

explanation can be given for this observation. Thus, further experiments will be necessary to understand this phenomenon.

**Author Contributions:** Conceptualization, methodology, validation and formal analysis, M.A and C.F.; investigation and data curation, M.A.; writing—original draft preparation, C.F.; writing—review and editing, M.A and C.F.; supervision, project administration, and funding acquisition, C.F. All authors have read and agreed to the published version of the manuscript.

**Funding:** This research was funded by French ANR agency under contract No. ANR-11-Labx-0005-01 CaPPA, the Région Hauts-de-France, the Ministère de l'Enseignement Supérieur et de la Recherche (CPER Climibio) and the European Fund for Regional Economic Development.

**Institutional Review Board Statement:** Not applicable.

**Informed Consent Statement:** Not applicable.

**Data Availability Statement:** Data can be obtained from the authors on request.

**Acknowledgments:** This project was supported by the French ANR agency under contract No. ANR-11-Labx-0005-01 CaPPA (Chemical and Physical Properties of the Atmosphere), the Région Hauts-de-France, the Ministère de l'Enseignement Supérieur et de la Recherche (CPER Climibio) and the European Fund for Regional Economic Development. The authors thank E. Assaf and M. Rolletter for help with initial measurements. This manuscript is dedicated to Robert Lesclaux.

**Conflicts of Interest:** The authors declare no conflict of interest.

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
