# Peer review of "Rate Constants and Branching Ratios for the Self-Reaction of Acetyl Peroxy (CH3C(O)O2) and Its Reaction with CH3O2"

_atmosphere, doi:10.3390/atmos13020186_

Round 1

Reviewer 1 Report

The manuscript presents new results on the rate constant and branching rations for self reaction of acetyl peroxy and its reaction with CH3O2. The manuscript can be published after minor revision. Further revision is not needed.

Commet :

  • The  work of Hui et al J. Phys. Chem. A 2019, 123, 49644972 concerning the reaction of Cl atom with CH3OH is not discussed.
  • Symbol of radical species ° is not used in this manuscript
  • Some of the rate constants are konw in solution and they can be also discussed and compared.

Author Response

  • The work of Hui et al J. Phys. Chem. A 2019, 123, 4964−4972 concerning the reaction of Cl atom with CH3OH is not discussed.

Indeed, this paper is not discussed, but the reaction of Cl-atoms with CH3OH does not play any role in our system, we never use CH3OH in this work, so we do not understand for what reason this paper should be discussed.

  • Symbol of radical species ° is not used in this manuscript

The radical symbol “°” has been added throughout the manuscript.

  • Some of the rate constants are known in solution and they can be also discussed and compared.

Sorry, but we could not find references for the title reactions in solution.

Reviewer 2 Report

Overall a good paper. One important change is required in the manuscript. The authors have used a bit informal writing style. For a research article, third person formal writing is required.

For example, Page 3, line 104: Manuscript reads "Here, we detect CH3C(O)O2 and CH3O2 radicals by absorption in the Ã-?̃ electronic transition, 
and the HO2 radical by absorption in the 21 overtone vibration band, all located in the near infrared region."

Using "we" kind of writing is not recommended. This sentence could read as " CH3C(O)O2 and CH3O2 radicals were detected by absorption in the Ã-?̃ electronic transition, and the HO2 radical by absorption in the 21 overtone vibration band, all located in the near infrared region".  

Suggest please review the entire manuscript and make changes to the sentences to conform to third person.

Introduction has enough references and is written well.

Author Response

Overall a good paper. One important change is required in the manuscript. The authors have used a bit informal writing style. For a research article, third person formal writing is required.

For example, Page 3, line 104: Manuscript reads "Here, we detect CH3C(O)O2 and CH3O2 radicals by absorption in the Ã-?̃ electronic transition, 
and the HO2 radical by absorption in the 2n1 overtone vibration band, all located in the near infrared region."

Using "we" kind of writing is not recommended. This sentence could read as " CH3C(O)O2 and CH3O2 radicals were detected by absorption in the Ã-?̃ electronic transition, and the HO2 radical by absorption in the 2n1 overtone vibration band, all located in the near infrared region".  

Suggest please review the entire manuscript and make changes to the sentences to conform to third person.

Introduction has enough references and is written well.

Thank you for the review. The language has been changed all over the manuscript.

Reviewer 3 Report

  1. The introduction part is too long with discussion concerning comparisons and disagreement with literature data. This information together with Table 1 should be moved to the discussion part. The authors should only emphasize the existing problem, disagreement of data and how they will solve this problem.
  2. The introduction part is missing important details about the method used (CRDS) and very similar applications in molecular trace gas detection, which can give a brief introduction to the readers. Some papers should be discussed in this part: (Photonics 2020, 7(3), 74; https://doi.org/10.3390/photonics7030074).
  1. The experimental part presented poor description of the characterization experimental procedure CRDS, the devices used are not described but only cited in other papers. It is important to report also the characteristics of lasers used.
  1. The change of ambient temperature will affect the cavity length, leading to the drift of resonant cavity mode. How this effect was controlled?
  2. Authors should give specification of the cavity mirrors reflectivity; the theoretical empty cavity ring down time should be calculated, and compared to the experimental value.
  3. The authors did not mention how they performed the data-acquisition. Whether they used any high speed digitizers for data sampling and analysis? If yes, the details of the digitizers need to mention. Please clearly mention which program language the authors used for the analysis of the data. If not, how the authors determine the averaging of the ring-down signals?
  4. The authors did not mention anything about stability of the system. The authors need to perform the Allan variance test to analyse the stability of the optical cavity and to determine the sensitivity of the system.
  5. The authors should mention about the precision/ accuracy and response time of the system during the measurement.
  6. From figures 3 to 10 the authors need to be more clear by used the format, Figure 3(a)…..3(e);……

Author Response

  1. The introduction part is too long with discussion concerning comparisons and disagreement with literature data. This information together with Table 1 should be moved to the discussion part. The authors should only emphasize the existing problem, disagreement of data and how they will solve this problem.

We do not agree with this point of view, because it seems important to us to explain in the introduction all the difficulties around the measurement of this reaction. For this, the different literature studies available need to be explained.

  1. The introduction part is missing important details about the method used (CRDS) and very similar applications in molecular trace gas detection, which can give a brief introduction to the readers. Some papers should be discussed in this part: (Photonics 2020, 7(3), 74; https://doi.org/10.3390/photonics7030074).

Again, we do not agree with the reviewer: cw-CRDS is nowadays a “standard” method for sensitive absorption measurements. Also, molecular trace gas detection is rather far away from the topic treated in this paper: here, only reactive radicals and not stable molecules are detected, and also measurements are done time-resolved. Therefore, a discussion on molecular trace gas detection by CRDS does not seem to us to be useful to the reader.

  1. The experimental part presented poor description of the characterization experimental procedure CRDS, the devices used are not described but only cited in other papers. It is important to report also the characteristics of lasers used.

We think that it is very common practice in the scientific literature to not describe in detail an experimental set-up anymore that has been used unchanged since many years and of which the details have been published in earlier papers. We have added the details on the DFB lasers in the manuscript, line 124:

(CH3C(O)O2°: Alcatel A1905LMI 3CN004 1 0CR, 6497±18 cm-1, HO2°: NEL NLK1E5GAAA, 6629±17 cm-1, CH3O2°: NEL NLK1B5EAAA, 7480±20 cm-1).  

  1. The change of ambient temperature will affect the cavity length, leading to the drift of resonant cavity mode. How this effect was controlled?

Common practice in cw-CRDS set-ups is to modulate the cavity length by mounting one of the cavity mirrors onto a piezo crystal. By modulating over at least one free spectral range, the cavity length comes periodically into resonance with the laser wavelength, and hence any drift due to change in ambient temperature doesn’t matter. Some more details have been added to the manuscript line 129.

One of the cavity mirrors is glued onto a piezo-transducer, which modulates periodically the cavity length in order to bring the cavity into resonance with the wavelength of the DFB lasers. The piezo is controlled by a homemade tracking system [37].

  1. Authors should give specification of the cavity mirrors reflectivity; the theoretical empty cavity ring down time should be calculated, and compared to the experimental value.

We do not have precise specifications of the mirrors: we use the same pairs since several years, and the reflectivity of course depends also on the cleanliness, which might change with time. On good days, we obtain ring-down times of the empty cavity of around 100 µs, which corresponds to a theoretical reflectivity of 0.99997. We have added line 143:

Typical ring-down times of the empty cavity were up to 100 µs, corresponding to reflectivity of the mirrors of 0.99997.

  1. The authors did not mention how they performed the data-acquisition. Whether they used any high speed digitizers for data sampling and analysis? If yes, the details of the digitizers need to mention. Please clearly mention which program language the authors used for the analysis of the data. If not, how the authors determine the averaging of the ring-down signals?

Details on the synchronisation and the data acquisition are given in earlier papers. We have added line 131 the following:

Then, the decay of light intensity is recorded by a fast 16-bit analogue acquisition card (PCI-6259, National Instruments) in a PC. The acquisition card has an acquisition frequency of 1.25 MHz and thus the ring-down signal is sampled every 800 ns and the data are transferred to PC in real time via PCI bus. An exponential fit is applied to retrieve the ring-down time. Through synchronisation with the trigger of the photolysis laser the delay between the photolysis pulse the random occurrence of the ring-down event is registered [36]. A typical kinetic decay is obtained by accumulating ring-down events for 50 to 100 photolysis pulses and consists of several hundreds of individual ring-down times that have occurred randomly either before or after the photolysis pulse.

In the legend of Figure 1 we have added:

Each data point results from one ring-down event, no averaging has been performed.

  1. The authors did not mention anything about stability of the system. The authors need to perform the Allan variance test to analyse the stability of the optical cavity and to determine the sensitivity of the system.

As we do not do continuous measurements of stable concentrations of a molecular trace gas, but time-resolved measurements of reactive species, it does not seem possible to us to perform Allan variance tests. Fluctuations in our system are also due to fluctuation in the photolysis energy (and with this the concentrations of the radicals), and this factor is more important than the stability of the optical cavity.

  1. The authors should mention about the precision/ accuracy and response time of the system during the measurement.

As can be seen from the different figures, the kinetic decays occur on the ms time-scale, while the ring-down times occur on the 10’s µs time scale. Therefore, we consider that we can safely treat the ring-down events as if the concentrations of reactive species are stable during the ring-down time. From this consideration we can say that the response time is limited by the ring-down time, i.e. kinetic decays should be much slower than the individual ring-down time. As no averaging is carried out, the accuracy of the measurement can be evaluated from the scatter of the individual ring-down times. As already mentioned above, the scatter in the energy of the photolysis laser is the most important limitation of the S/N ratio.

  1. From figures 3 to 10 the authors need to be more clear by used the format, Figure 3(a)…..3(e);……

Letters have been added to the individual figures and the text has been changed accordingly.

Reviewer 4 Report

This work reports an experimental estimation of rate constants and the product branching ratios for the self-reaction of acetyl peroxy and its reaction with CH3O2. This work is well-written and rigorous, so I recommend publication in Atmosphere. I only have a minor comment.

I recommend that authors compare the thermal rate constant estimated in their study with previous theoretical studies (if available). If possible,  DFT calculations combined with the TST approach can be applied to estimate the thermal rate constants. I believe that the theoretical estimations can corroborate the experimental findings.

Author Response

This work reports an experimental estimation of rate constants and the product branching ratios for the self-reaction of acetyl peroxy and its reaction with CH3O2. This work is well-written and rigorous, so I recommend publication in Atmosphere. I only have a minor comment.

I recommend that authors compare the thermal rate constant estimated in their study with previous theoretical studies (if available). If possible, DFT calculations combined with the TST approach can be applied to estimate the thermal rate constants. I believe that the theoretical estimations can corroborate the experimental findings.

Thank you for your review. To our knowledge, no theoretical studies have been carried out on the title reaction, only the reaction between CH3C(O)O2 and HO2 has been studied theoretically. The authors agree that a theoretical study would be very interesting, especially to investigate the branching ratio of the cross reaction between CH3C(O)O2 and CH3O2, but this is beyond our possibilities and could be carried out in the future by theoreticians. However, we have added line 98 a reference for a semi-empirical study of (R1) and (R2).

A semi-empirical study on the rate constants of self- and cross-reactions of peroxy radicals, based on the calculated stabilisation energy of the tetroxide intermediate, predicts 1.4×10-11 cm3s-1 for (R1) and 7×10-12 cm3s-1 for (R2), which is in good agreement with experiments for (R1) and at the lower end for (R2) [31].

In the discussion, line 459, we have added:

A comparison with the predictions of the semi-empirical study [31] is not shown, because no branching ratios are predicted, which is indispensable for the prediction of concentration time-profiles. However, the rate constant for (R2) predicted in the semi-empirical study, based on the stabilization energy of the tetroxide intermediate, is with 7×10-12 cm3s-1 even below the lowest experimental value and it can therefore be supposed, that the semi-empirical method is not reliable for this type of cross-reactions. To our knowledge, no theoretical calculations concerning mechanism and rate constants of (R1) and (R2), and more important of the branching ratio for (R2), have been carried out, but would certainly be interesting.

Round 2

Reviewer 3 Report

It can be accepted in the present form